# Evaluation of Hydroxyl Radical Reactivity by Thioether Group Proximity in Model Peptide Backbone: Methionine versus S-Methyl-Cysteine

**DOI:** 10.3390/ijms23126550

**Published:** 2022-06-11

**Authors:** Chryssostomos Chatgilialoglu, Magdalena Grzelak, Konrad Skotnicki, Piotr Filipiak, Franciszek Kazmierczak, Gordon L. Hug, Krzysztof Bobrowski, Bronislaw Marciniak

**Affiliations:** 1Center for Advanced Technology, Adam Mickiewicz University, Uniwersytetu Poznanskiego 10, 61-614 Poznan, Poland; chrys@isof.cnr.it (C.C.); magdalena.grzelak@amu.edu.pl (M.G.); piotrf@amu.edu.pl (P.F.); 2ISOF, Consiglio Nazionale delle Ricerche, Via P. Gobetti 101, 40129 Bologna, Italy; 3Institute of Nuclear Chemistry and Technology, Dorodna 16, 03-195 Warsaw, Poland; k.skotnicki@ichtj.waw.pl; 4Faculty of Chemistry, Adam Mickiewicz University, Uniwersytetu Poznanskiego 8, 61-614 Poznan, Poland; kazmak@amu.edu.pl; 5Radiation Laboratory, University of Notre Dame, Notre Dame, IN 46556, USA; hug.1@nd.edu

**Keywords:** methionine, S-methyl-cysteine, pulse and γ-radiolysis, oxidation, free radicals, high-resolution MS/MS

## Abstract

Hydroxyl radicals (HO^•^) have long been regarded as a major source of cellular damage. The reaction of HO^•^ with methionine residues (Met) in peptides and proteins is a complex multistep process. Although the reaction mechanism has been intensively studied, some essential parts remain unsolved. In the present study we examined the reaction of HO^•^ generated by ionizing radiation in aqueous solutions under anoxic conditions with two compounds representing the simplest model peptide backbone CH_3_C(O)NHCHXC(O)NHCH_3_, where X = CH_2_CH_2_SCH_3_ or CH_2_SCH_3_, i.e., the Met derivative in comparison with the cysteine-methylated derivative. We performed the identification and quantification of transient species by pulse radiolysis and final products by LC–MS and high-resolution MS/MS after γ-radiolysis. The results allowed us to draw for each compound a mechanistic scheme. The fate of the initial one-electron oxidation at the sulfur atom depends on its distance from the peptide backbone and involves transient species of five-membered and/or six-membered ring formations with different heteroatoms present in the backbone as well as quite different rates of deprotonation in forming α-(alkylthio)alkyl radicals.

## 1. Introduction

The reactive oxygen species (ROS) network, initiated from a superoxide radical anion (O_2_^•−^) and nitric oxide (NO^•^), regulates numerous metabolic processes, although the overproduction of ROS has been linked with the etiology of various diseases [1,2]. Hydroxyl radicals (HO^•^) are the most reactive species within this network and have long been regarded as a major source of cellular damage [3]. The main cellular processes that generate HO^•^ are depicted in Figure 1 [4]. H_2_O_2_ is at the crossroad of several pathways of HO^•^ formation, the main ones being by the Fenton reaction (Fe^2+^ and H_2_O_2_), the Haber–Weiss reaction (O_2_^•−^ and H_2_O_2_), and the reduction of hypochlorous acid (HOCl) by O_2_^•−^. The spontaneous decomposition of ONOOH is an additional pathway [5]. The diffusion distance of HO^•^ is very small because of its high reactivity with all types of biomolecules and, consequently, there is a low probability to be intercepted by antioxidants [5,6]. In this context, aerobic life would not be possible without the enzymes superoxide dismutase (SOD) and catalase (CAT) that transform superoxide to water and oxygen (Figure 1) [3].

The one-electron and two-electron oxidation of sulfide moieties have been intensively studied by a variety of methods, going from organic chemistry to biology and industrial processes. The oxidation of methionine (Met) is an important reaction that plays a key role in protein modifications during oxidative stress and aging [7]. Figure 2 shows the reaction of Met residues with two-electron oxidants such as H_2_O_2_, ONOO^−^, or HOCl, with the formation of methionine sulfoxide, Met(O) [7], whereas the reaction of HO^•^ with Met gives formally one-electron oxidation through two consecutive steps, i.e., the formation of an adduct radical followed by heterolytic cleavage [8]. 

The radiolysis of water provides a very convenient pathway for generating hydroxyl radicals (HO^•^). Time-resolved kinetic studies by pulse radiolysis have expanded our mechanistic understanding of radical reactivity of Met in differently functionalized environments [8]. An overview of the transient-species detection in the one-electron oxidation of Met derivatives by various time-resolved techniques has been recently presented [9]. Mechanistic aspects of Met oxidation according to various structural characteristics given by peptide sequences and pH by one-electron oxidants (including HO^•^) are summarized and discussed. Neighboring group participation seems to be an essential interaction that controls the outcome of the one-electron oxidation of methionine. The observed transient species are precursors of the final products [9]. On the other hand, the photosensitized oxidation of Met derivatives by the excited triplet of 3-carboxybenzophenone was shown to differ significantly, leading to α-thioalkyl radical formation through an electron transfer coupled by proton transfer (ET–PT) within the encounter complex [10].

There are also a few ionizing radiation chemical studies of Met in aqueous solutions, followed by product characterization and quantification. We recently described the reaction of HO^•^ with Ac-Met-OMe in aqueous solutions under anoxic conditions, performing identification and quantification of transient species by pulse radiolysis and the final products by LC–MS and high-resolution MS/MS after γ-radiolysis [11]. It was demonstrated that the formation of small amounts of the corresponding sulfoxide is due to in situ formation of H_2_O_2_ rather than to direct oxidation by HO^•^ [11]. Additionally, the reaction of HO^•^ with the tripeptide Gly-Met-Gly provided strong evidence that the corresponding Met sulfoxide in the tripeptide derives from the in situ formed H_2_O_2_ [12]. 

Based on these premises, here we report the study on the reactions of HO^•^ with methionine (Met) derivative **1**, and with its congener cysteine-methylated (Cys-Me) derivative **2** under anoxic conditions (Figure 3). Both of them contain the same model peptide backbone: compound **1** represents the simplest case of a Met residue located in a peptide sequence, at the same time eliminating the known chemistry of unsubstituted Met, i.e., occurrence of intramolecular proton transfer from the amino group and decarboxylation [9,13]. The current reactivity comparison between compounds **1** and **2** aims at understanding the influence of the thioether group distance from the peptide backbone (**2** has one CH_2_ less) on the chemistry of HO^•^ radical. In particular, we provided identification and quantification of transient species by pulse radiolysis and final products of γ-radiolysis by LC–MS and high-resolution MS/MS. 

## 2. Results and Discussion

Ionizing radiation of neutral water leads to the primary reactive species e^−^_aq_, HO^•^, and H^•^, as shown in Reaction one. The values in brackets represent the radiation chemical yield (*G*) in µmol J^−1^ [14]. In N_2_O-saturated solution (~0.02 M of N_2_O), e^−^_aq_ are efficiently transformed into HO^•^ radicals via Reaction two (*k* = 9.1 × 10^9^ M^−1^ s^−1^) [15], affording *G*(HO^•^) = 0.56 µmol J^−1^.
H_2_O 
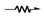
 e^−^_aq_ (0.28), HO^•^ (0.28), H^•^ (0.06)(1)
e^−^_aq_ + N_2_O + H_2_O → HO^•^ + N_2_ + HO^−^
(2)

Thus in these experimantal conditions HO^•^ radicals were the main oxidizing species. 

### 2.1. Pulse Radiolysis, Spectra Resolution, and Concentration Profiles of Transients

The reaction of HO^●^ with compounds **1** and **2** was investigated in N_2_O-saturated solutions of 0.2 mM **1** or **2**, at natural pH (pH value of 7.0 was recorded) to mimic natural biological conditions. Transient absorption spectra in the range 270–700 nm recorded in the time range of 1 μs to 160 μs are presented in Figure 1. 

The transient spectrum obtained 1 μs after the electron pulse in N_2_O-saturated aqueous solutions containing 0.2 mM of compound **1** at pH 7 and showed a dominant absorption band with λ_max_ = 330 nm (Figure 1A). Based on our previous studies on dimethyl sulfide [16], and various Met-containing derivatives [11,12,17,18], this band can be assigned unequivocally to the HO**^•^** adduct at the sulfur atom **HOS**^•^, established as the first step in the HO**^•^**-induced oxidation of compound **1** (see Figure 4). However, as shown in Figure 1A, the observed transient-spectral profile changed with time, indicating contributions from various transient species. The spectrum obtained 1 µs after the electron pulse underwent further changes and, at 3 µs after the electron pulse, had transformed into the spectrum showing two pronounced shoulders in the 380–400 nm and 450–500 nm ranges. The latter shoulder further developed into an absorption band with λ_max_ = 470 nm, which was clearly seen in the absorption spectrum obtained 10 μs after the pulse together with an absorption band with a maximum located around λ = 290 nm. These absorption bands reached their maximum absorbance values around 7 and 15 μs after the electron pulse, respectively (see upper inset in Figure 1A). These absorption bands can be tentatively assigned to the intermolecular sulfur–sulfur three-electron bonded radical cations (**SS^•+^**) and α-(alkylthio)alkyl radicals (**αS(1)^•^** and/or **αS(2)^•^**), respectively (Figure 4). Moreover, a distinct flat region between 320–400 nm was observed indicating the presence of various additional transient species. Based on our previous studies on Ac-Met-OMe [11], the C_α_-centered radicals (**αC(1)^•^** and/or **αC(2)^•^**) and intramolecular sulfur–nitrogen three-electron bonded radicals (**SN(1)^•^** and/or **SN(2)^•^**) can be considered as potential transients (Figure 4). It is noteworthy that the transient-absorption spectrum obtained 30 μs after the pulse is characterized by the distinct absorption band with λ_max_ = 290 nm and an even more pronounced flat region between 320–460 nm, which can be rationalized by the faster decay kinetics of **SS^•+^** compared to the decay kinetics of the remaining transients, i.e., **αS^•^** and **SN^•^.** This fact is clearly confirmed by a comparison of the short- and long-time kinetic profiles recorded at four selected wavelengths (290, 340, 390, and 490 nm) corresponding to the absorption maxima of the four most abundant radicals present in the irradiated system, i.e., **αS^•^**, **HOS^•^**, **SN^•^**, and **SS^•+^** (see insets in Figure 1A). The short-time kinetic traces look different, and they reached their maximum absorbance signal at various times after the pulse. Moreover, the long-time kinetic profiles in the time domain above 70 μs clearly illustrate the kinetics of radical–radical coupling reactions (cf. lower inset in Figure 1A).

In order to check whether the thioether group distance from the peptide backbone has an influence on the character of intermediates resulting from the reactions of HO^•^ radicals, transient absorption spectra were recorded after the electron pulse in N_2_O-saturated aqueous solutions containing 0.2 mM **2** at pH 7. The transient absorption spectra recorded in the time domain between 1 and 10 μs after the electron pulse are characterized by different features in comparison to the absorption spectra recorded in the case of **1** (cf. Figure 1A,B). They consisted of two absorption bands with maxima located at λ = 290 and 400 nm. The formation of these bands was fully developed within the range of 3 μs. However, the kinetic time profiles recorded in the time domain >3 μs at 290 nm and 400 nm were different (Figure 1B, upper insert). While the absorption at 400 nm started to decay, the absorption at 290 nm was stable. Consequently, at 3 μs the absorption band with λ_max_ = 400 nm was more intense than the absorption band with λ_max_ = 290 nm, whereas at 10 μs the intensity of these absorption bands reversed. These absorption bands can be tentatively assigned to α-(alkylthio)alkyl radicals (**αS**^•^**)** and intramolecular sulfur–oxygen three-electron bonded radical cations (**SO**^•**+**^) (Figure 5). Interestingly, no distinct absorption bands with λ_max_ = 330 and 470 nm were observed, which could be assigned to **HOS**^•^ and **SS**^•**+**^, respectively (Figure 1B). With the lapse of time to when the 400 nm absorption band disappeared completely (Figure 1B, lower inset), the absorption spectrum recorded after 160 μs is characterized only by a pronounced absorption band with λ_max_ = 290 nm.

The purpose of this subsection is to provide arguments, based on spectral and kinetic analyses, for the presence of potential transients, which are formed during the HO^•^-induced oxidation of **1** and **2** in aqueous solutions. Resolution of the absorption spectra recorded at any desired time delay, following the electron pulse ranging from 200 ns to 80 μs, into contributions from these transients allows for the quantification (as *G*-values) and extraction of their concentration profiles. This is a key issue that enables justification for the mechanistic assumptions reported in Figure 4 and Figure 5, also explaining the formation of stable products (see Section 2.2).

Based on our previous work on Ac-Met-OMe [11], the following transient species **HOS**^•^, **αC(1)**^•^, **αS(1)**^•^, **αS(2)**^•^, **SS**^•**+**^, **SN(1)**^•^, and **SN(2)**^•^ in Figure 4 were taken into account in spectral resolutions (see also Appendix A). Since the amide nitrogen atom is much more effective in activating the methyl group for H-atom abstraction than the oxygen atom in the ester group [19], the additional C-centered radical **αC(2)**^•^ was also considered in these spectral resolutions. The transient absorption spectrum recorded 1 μs after the electron pulse was resolved into contributions from the three components **HOS**^•^, **αC(1)**^•^, and **SS**^•**+**^ with the respective *G*-values 0.27, 0.08, and 0.02 µmol J^−1^ that correspond to 73%, 22%, and 5% of all radicals present at that time (Table 1). The calculated total *G*-value of 0.37 μmol J^−1^ does not constitute the expected *G*-value of HO^•^ (0.56 μmol J^−1^) available for the reaction of **1** at pH 7 and the concentration (0.2 mM) of **1**. This difference in *G*-values can be easily understood since at this time, the reactions of HO^•^ with **1** via all pathways presented at Figure 4 are not completed yet. However, their percentage contributions confirm the domination of the **HOS**^•^ in the transient absorption spectrum and rationalizes its spectral features (cf. Figure 1A). From the concentration profiles obtained from the resolution of the spectra recorded at various times after the electron pulse, one can see that at 1.4 μs after the pulse **HOS**^•^ reaches its maximum *G*-value = 0.33 μmol J^−1^ (Figure 2A). It is still the most abundant radical in the irradiated solution containing **1**, and its concentration constitutes 72% of all radicals present at that time (Table 1). However, there are additional spectral components that start to contribute to this spectrum and include the following radicals: **αC(1)**^•^, **αS**^•^, **SS**^•**+**^, and **SN**^•^ with the respective *G*-values: 0.08, 0.01, 0.03, and 0.01 μmol J^−1^ (Table 1). These intermediates were previously identified during HO^•^-induced oxidation of Ac–Met–OMe under similar experimental conditions [11]. The calculated total *G*-value of 0.46 μmol J^−1^ does not constitute yet the expected *G*-value of HO**^•^** radicals available for the reaction of **1**. This difference in *G*-values can be rationalized similarly as for the earlier recording time equal to 1 μs. However, the appearance of **SS^•+^** and **SN^•^** also indicates an involvement of secondary reactions with the participation of **HOS^•^** (Figure 4).

Using the same spectral resolution approach, the transient spectrum recorded 4 µs after the electron pulse was resolved into contributions from the same components, however, resulting in different fitted *G*-values: 0.12, 0.09, 0.14, 0.10, and 0.08 μmol J^−1^, respectively (Table 1), and additionally from **αC(2)^●^** with the *G*-value = 0.03. The total *G*-value of all radicals present (0.56 μmol J^−1^) is in excellent agreement with the expected yield of HO^•^ (vide supra). This is the time point when all HO^●^ have reacted with **1**, but not all **HOS**^•^ have yet undergone transformations into the secondary transients.

The next key time point is 10 µs after the electron pulse when all **HOS^●^** have undergone full transformations into the secondary products. The spectrum was resolved into contributions from the species **αC(1)**^•^, **αC(2)**^•^, **αS**^•^, **SS**^•**+**^, and **SN**^•^, with fitted *G*-values: 0.07, 0.08, 0.23, 0.12, and 0.10 μmol J^−1^, respectively (Table 1). The sum of all component spectra with their respective radiation chemical yields (*G*-values) resulted in a good fit (⊗ symbols in inset of Figure 2A) to the experimental spectrum. Their percentage contributions to the transient absorption spectrum rationalizes nicely its spectral features (Figure 1A). The total *G*-value of all radicals present (0.60 μmol J^−1^) slightly exceeds the expected yield of HO^•^ (vide supra). However, this can be rationalized by the fact that H-atoms formed during radiolysis of water (Equation (1)) can also form **αC(1)**^•^, **αC(2)**^•^, **αS(1)**^•^, and **αS(2)**^•^ by direct H-atom abstraction from **1**. It is worthy to note that the most abundant radicals present at this time are **αS(1)**^•^ and **αS(2)**^•^, which constitute nearly 40% of all radicals (Table 1). The excellent material balance of all radicals identified in the system equal to the *G*-value of HO^●^ available for the reaction with **1** proves that they are precursors of the corresponding end-products. At this point, it has to be stressed that after 10 μs, following the electron pulse, the total *G*-values of the radicals began to decrease slowly, reaching the value of 0.54, 0.44, and 0.37 μmol J^−1^ at 30, 80, and 170 μs, respectively (Table 1). This suggests that in this time domain 40% of all of the radicals formed in the system were consumed in radical–radical processes leading to the final products. Moreover, over this time domain **αS(1)^●^** and **αS(2)^●^** remain the most abundant radicals, constituting nearly 50% of all of the radicals on the timescale of hundreds of microseconds.

Based on our previous work on S-methyl-glutathione (a tripeptide containing an internal S-methyl-cysteine residue) [20], the following species of **HOS^●^**, **αS^●^**, **SS^●+^**, and **SO^●+^** were taken into account in spectral resolutions (Figure 5 and Appendix A). Additional support for taking intramolecular sulfur–oxygen three-electron-bonded radicals (**SO^●+^**) into the spectral resolutions came from the photosensitized oxidation studies of **two** where **SO^●+^** was identified as one of the transient products [10]. Since the amide nitrogen atom is much more effective in activating the methyl group for H-atom abstraction than the oxygen atom of the ester group [19], the **αC(2)^●^** was also taken into account in the current spectral resolutions.

The first significant difference between compounds **1** and **2** concerns the concentration profiles of **HOS^•^** radicals. In solutions containing **2**, **HOS^•^** reach their maximum *G*-value = 0.16 μmol J^−1^ at 0.5 μs after the electron pulse, compared to 1.4 μs for **1** (cf. Figure 2A,B). Moreover, this maximum *G*-value is slightly over two-fold lower than that observed for **1**. Simultaneously, at the same time a comparable *G*-value = 0.12 μmol J^−1^ for **SO^•+^** was obtained (Table 2). This substantial decrease in this **HOS^•^**
*G*-value can be rationalized by the occurrence of an additional very fast **HOS^•^** decay process leading to **SO^•+^** formation at the expense of the measured *G*(**HOS^•^**). This decay channel is not effective in **1.** In principle, **HOS^•^** has been shown to decay along three different pathways: (i) a spontaneous unimolecular dissociation into **S^•+^** and HO^−^, (ii) a proton-assisted elimination of HO^−^ leading to **S^•+^** and H_2_O, and (iii) a displacement of HO^−^ by a second thioether molecule leading to **SS^•+^**. Typical values of the rate constants for these processes are: (5–7) × 10^5^ s^−1^, 1 × 10^10^ M^−1^ s^−1^ and 1 × 10^8^ M^−1^ s^−1^, respectively. Since the experimental conditions (pH = 7 and concentration of **1** and **2** = 0.2 mM) were similar, these three reactions should occur with the similar rates for both compounds. Moreover, taking into account the experimental conditions, the last two reaction pathways responsible for the decay of **HOS^•^** are negligible for both compounds **1** and **2**. At this point the following question arises: what is the nature of this additional process present in **2** that can compete with the spontaneous dissociation of **HOS^•^** (Figure 5)? Based on the known radical chemistry coming from research on the oxidation of (carboxyalkyl)thiopropionic acid derivatives [21] and β-, γ-, and δ-hydroxyalkyl sulfides [22], an intramolecular carbonyl-assisted decay of **HOS^•^** radicals can occur that directly leads to the five-membered **SO^•+^** intermediate (Figure 5).

Compatible with the above observations is the time point equal to 2.25 μs after the electron pulse (Table 2) when all HO**^•^** radicals have reacted with **2**, but not all **HOS****^•^** have yet undergone transformations into the secondary products. The transient spectrum was resolved into contributions from the four components: **HOS****^•^**, **αC(2)****^•^**, **αS****^•^**, and **SO****^•+^** using the respective fitted *G*-values: 0.10, 0.03, 0.16, and 0.27 μmol J^−1^. Moreover, the most abundant transient species at that time is **SO****^•+^** contrary to the system containing **1**, where **HOS****^•^** was still the most abundant radical. For comparison, this specific time point for **1** was equal to 4 μs (Table 1 for 4 μs). 

The next key time point is 10 µs after the electron pulse (similar as for 1) when all **HOS****^•^** have undergone full transformations into the secondary products. The spectrum was resolved into contributions from the following components **αC(2)**^•^, **αS****^•^**, **SS****^•+^**, and **SO****^•+^** (vide inset in Figure 2B), with the following fitted *G*-values: 0.15, 0.23, 0.01, and 0.22 μmol J^−1^, respectively (Table 2). The sum of all component spectra with their respective radiation chemical yields (*G*-values) resulted in a good fit (⊗ symbols in inset of Figure 2B) to the experimental spectrum. The total *G*-value of all transient species present (0.61 μmol J^−1^) can be similarly rationalized as for **1** (vide supra). The excellent material balance, that all of the radicals identified in the system are equal to the *G*-value of HO^•^ available for the reaction with **2**, proves that the identified radical transients are precursors of the corresponding end products. At this point, it has to be emphasized that, at 10 μs after the electron pulse, the **αC(2)**^•^ radicals reached their highest *G*-values for both compounds **1** and **2** equal to 0.07 and 0.15 μmol J^−1^, respectively (cf. Table 1 and Table 2). This is the second feature differentiating these compounds. Therefore, the next questions arise: what is the reason for the observed differences in the *G*-values of **αC(2)**^•^ formed in **1** and **2**, and why was the formation of **αC(2)**^•^ more efficient in **2**? 

In order to address these two questions, we propose the following explanation. The **αC(2)**^•^ radicals in **1** and **2** are formed via two pathways: via direct H-atom abstraction from the methyl group attached to the amide nitrogen atom by HO^•^, or indirectly by a sequence of reactions involving **SN(2)**^•^ in the case of **1**, and **SO**^•**+**^ in the case of **2** as precursors of **αC(2)**^•^ (cf. Figure 4 and Figure 5). The rate constants of all primary reactions of HO^•^ radicals with **1** and **2** leading to **HOS**^•^, **αS(1)**^•^ and **αS(2)**^•^, **αC(1)**^•^, and **αC(2)**^•^ are similar for both compounds. Therefore, the sequences of reactions leading to **SN(2)**^•^ for **1** and **SO**^•**+**^ for **2** seem to be responsible for the observed differences in radiation chemical yields (*G*-values) of **αC(2)**^•^ radicals.

In the case of compound **1**, the common precursor for **SN(1)**^•^, **SN(2)**^•^, **αS(1)**^•^, and **αS(2)**^•^ is the monomeric sulfur radical cation (**S**^•**+**^), which is formed directly from **HOS**^•^**.** Comparing the *G*-values of transients at 10 μs (when all **HOS**^•^ have undergone full transformations into the secondary products), it is clearly visible that **αS^•^** are formed with the largest efficiency, which means that the deprotonation of **S^•+^** is more efficient than the formation of **SN^•^** (Table 1). Moreover, the formation of the six-membered **SN(2)^•^** is kinetically less favorable than the formation of the five-membered **SN(1)^•^**. As a consequence, the smallest amount of **S^●+^** can be converted to **SN(2)^•^**. It has to be also noted that not all **SN(2)^●^** radicals can be quantitatively converted into **αC(2)^•^** since **SN(2)^•^** exists in an equilibrium with **S^•+^** (cf. Figure 1). In the case of **2**, the formation of **SO^•+^** does not exclusively occur via **S^•+^**. A substantial amount of five-membered **SO^•+^** is formed directly from **HOS^•^**, which is the reason for a more efficient formation of **αC(2)^•^** in **2** (Figure 5). Contrary to **1**, at 10 μs the *G*-value of a direct precursor of **αC(2)^•^** in **2** is comparable with the *G*-value of **αS(1)^•^** and **αS(2)^•^** (cf. Table 1 and Table 2).

The third difference between these two compounds that needs to be explained is the negligible **SS^•^****^+^** contribution in transient absorption spectra recorded in solutions containing **2** (vide Figure 1B and Table 2). There are several possible scenarios that can be taken into account. The formation of **SO^•+^** directly from **HOS^•^**, excluding **S^•+^**, which is also a direct precursor of **SS^•+^**, can only partly explain this observation. Furthermore, based on our previous studies on S-methylglutathione [20], where the rate constant of fast formation of five-membered **SO^•+^** directly from **S^•+^** was estimated with the lower limit as >6 × 10^7^ s^−1^, the slow formation of **SS^•^****^+^** (taking into account the low concentration of **2**) is the next possible rationalization. In the case of **1**, the rate constant of five-membered **SN(1)^•^** formation directly from **S^•+^** was not measured earlier, however, it must be lower in comparison to the rate constant of the analogous reaction leading to **SO^•+^** (vide supra). 

The fourth difference between **1** and **2** is the lack of **αC(1)^•^** formation in the case of **2** (vide Table 2). This observation can be rationalized by the unimportance of the SN^●^ formation, in particular of SN(1)^●^, due to its unstable four-membered structure. Moreover, an additional possible potential reaction pathway leading to **αC(1)^•^** formation via direct H-abstraction seems to be also unimportant. The lack of **αC(1)^•^** formation was further confirmed by the lack of final products involving their contribution (cf. Table 3 and Table 4) (vide discussion in 2.2 addressing the issue of **αC(1)^•^** absence in **2**).

### 2.2. γ-Radiolysis and Product Analysis

In addition to the reactive species e_aq_^−^, HO^•^, and H^•^, the radiolysis of neutral water leads also to H^+^ (0.28) and H_2_O_2_ (0.07); in parenthesis: *G* in µmol J^−1^ [23]. In N_2_O-saturated solution, the *G*(HO^•^) = 0.56 µmol J^−1^, therefore HO^•^ and H^•^ account for 90% and 10%, respectively, of the reactive species (cf. Reactions (1) and (2)).

N_2_O-saturated solutions containing compounds **1** or **2** (1.0 mM) at natural pH were irradiated for 800 Gy under stationary state conditions with a dose rate of 46.7 Gy min^−1^ followed by LC–MS and high-resolution MS/MS analysis. Representative HPLC runs of irradiated samples are shown in Figure 3 and Figure 4 for compounds **1** and **2**, respectively.

It is well documented from previous studies on Met derivatives that the sulfoxide **3** is formed from the in-situ generation of hydrogen peroxide [11,12], whereas the H^•^ addition to the sulfur center (*k* = 1.7 × 10^9^ M^−1^ s^−1^) with the formation of a sulfuranyl radical intermediate affords compound **4** and the CH_3_S^•^ radical (Figure 6) [24,25]. In analogy, we expected the same reactions with starting material **2** giving **5** and **6**, respectively. It is worth underlining that these reactions are not detectable in the pulse-radiolysis study, therefore their existence is based on the final-product identifications.

Figure 3 shows the HPLC chromatogram obtained by the injection of the crude reaction mixture of compound **1** with a separation of 24 peaks including the starting material **1** (peak 3), whereas Figure 4 shows the chromatogram obtained from the reaction of compound **2** that contains 15 peaks including the starting material **2** (peak 3). All peaks were identified, and their chemical structures assigned by examination of their high-resolution mass data and characteristic fragmentation patterns (see Appendix A). In both cases, to the left of the starting material **1** or **2** we could identify the presence of sulfoxides **3** or **5** (peak 1) and the desulfurization compounds **4** or **6** (peak 2), respectively (cf. Appendix A). The desulfurization process with the formation of the thiyl radical (CH_3_S^•^) depicted in Figure 6 is extremely important, because some of the products derive from the combination of CH_3_S^•^ with carbon-centered radicals.

From the pulse radiolysis studies described above, the reactions of HO^•^ with **1** or **2** were depicted following several paths. In Figure 4 and Figure 5 the formation of the adduct radical (**HOS^•^**) is the main path, whereas four minor ones were identified as follows: the H-atom abstraction from the CH_2_-S-CH_3_ moiety to give the two **αS^•^** radicals and the H-atom abstraction from the N-CH-CO or NHCH_3_ moiety to give the two **αC^•^** radicals. The **HOS^•^** follows a first-order decay (*k*_d_ = 5.6 x 10^5^ s^−1^) by HO^−^ elimination to give the sulfide radical cation, which is at the crossroad of various possible reactions affording the intermediates **SS^•+^**, **αS^•^**, **SN^•^**, and **αC^•^** for compound **1**, and **SS^•+^**, **αS^•^**, **SO^•+^**, and **αC^•^** for compound **2**.

We suggest that in both starting materials the formation of the disulfide radical cation (**SS^•+^**), which is in equilibrium with the sulfide radical cation (S^•+^) and the starting material, is followed by the fragmentation of **SS^•+^** to afford the observed disulfide **7** or **8** (Figure 7, cf. Appendix A). Analogous products are reported in radiolytic studies of other methionine derivatives [10,11]. Such an asymmetrical disulfide was found to be the major product in the tripeptide Gly-Met-Gly transformation [12].

All the remaining peaks in both cases derived from the combination of two radicals. Table 3 and Table 4 collect the product formations as well as the precursor radicals in each case. The reaction of CH_3_S^•^, generated from the reaction of H^•^ with the substrates, can combine with various carbon-centered radicals to give the corresponding sulfides. Indeed, CH_3_S^•^ reacts with **αS(1)^•^**, **αS(2)^•^**, **αC(1)^•^**, and **αC(2)^•^** for compound **1** (Table 3) and with **αS(1)^•^**, **αS(2)^•^**, and **αC(2)^•^** for compound **2** (Table 4). Both tables report also the relative ratios based on the intensity of the peaks in the chromatograms. We suggest that the CH_3_S^•^ adduct is like a footprint of the relative concentrations of the four carbon-centered radicals. Based on this rationalization, the relative concentrations of **αS(2)^•^**, **αS(1)^•^**, **αC(1)^•^**, and **αC(2)^•^** are 15.4/9.4/1.4/1.0 for the methionine derivative (Table 3) whereas the relative concentrations of **αS(2)^•^**, **αS(1)^•^**, and **αC(2)^•^** are 72.2/1.0/2.3 for the cysteine-methylated derivative (Table 4). The absence of **αC(1)^•^** in the pulse radiolysis simulation of the transients derived from compound **2** can be attributed to fast and well documented β-fragmentation of thiyl radical (Figure 8). Since no evidence of formation of compound **9** is obtained by LC–MS and high-resolution MS/MS analysis, we concluded that this path is unimportant. This confirms the unimportance of SN^•^ formation in the radiation chemistry of compound **2** (cf. Figure 4 and Figure 5). Although **αS^•^** radicals were the most abundant in both experiments (**αC^•^** was at least an order of magnitude smaller than **αS^•^**), the relative percentage of the two isomers changed profoundly going from compound **1** to **2**, i.e., the ratio **αS(2)^•^**/**αS(1)^•^** is 15.4/9.4 and 72.2/1.0, respectively. 

It is expected for an **αS(2)^•^** radical to be found at a higher concentration than **αS(1)^•^** and in line with the higher stability of secondary vs. primary alkyl radical due to favorable deprotonation from the precursor sulfide radical cation. In the case of compound **1**, the analysis of high-resolution MS/MS spectra was in favor of **αS(2)^•^**/**αS(1)^•^** by 15.4/9.4 (see Appendix A). For compound **2** on the other hand, the major peak fits very well with both αS(2)—SCH_3_ and αS(1)—SCH_3_ (see Appendix A); since the number of observed diastereoisomers fits only with higher concentrations of **αS(2)^•^** radicals, we assigned the ratio 72.2/1.0 to **αS(2)^•^**/**αS(1)^•^**.

It has to be stressed that the above observations cannot be directly confirmed by the pulse-radiolysis study described above. It was only seen that **αS^•^** are formed with slightly different efficiencies in compounds **1** and **2** (cf. 0.28 μM vs. 0.23 μM, respectively). Unfortunately, it was not possible to distinguish between the two **αS(2)^•^** and **αS(1)^•^** radicals and as a consequence to obtain directly their respective *G*-values in both compounds and to extract directly the kinetics of their formation.

However, these interesting observations need some comments and explanations. Since the main source of **αS^•^** is the α-deprotonation of **S^●+^** [26], it seems obvious that the deprotonation of **S^•+^** to form **αS(2)^•^** is much faster for **2** than for **1**. As it was stated earlier [27], the efficiency of α-deprotonation strongly depends on the molecular structure of S^•+^ and, to be more specific, on the relative alignment of the singly occupied sulfur p-orbital with the C−H σ-bond to be cleaved. This fact was nicely illustrated by deprotonation kinetics on the example of the monomeric sulfur radical cations derived from di-isopropyl sulfide (*i*-(C_3_H_7_)_2_S) and from 9-thia-bicyclo-[3.3.1]-nonane. In the first case a high probability of parallel alignment of the sulfur p-orbital and the C−H σ-orbitals, facilitated by the steric influence of the two methyl groups in combination with the free rotation around the C–S bond, accelerates α-deprotonation of *i*-(C_3_H_7_)_2_S^●+^. On the other hand, in second case, the neighboring C−H bonds are fixed in a configuration perpendicular to the sulfur p-orbital, which practically prevents deprotonation [2]. Perhaps, for compound **2** exists the more favorable conformational arrangement in the **S^●+^**, than for compound **1,** for deprotonation leading to **α-S(2)^•^.** This can be facilitated, similarly as in *i*-(C_3_H_7_)_2_S^●+^, by the steric influence of the peptide backbone that is located closer in **2** affording a better parallel alignment of the sulfur p-orbital and the C−H σ-orbitals in the CH_2_ group in comparison to **1**.

The data from the high-resolution MS/MS spectra showed 16 dimeric compounds that are derived from the radiolytic study of starting material **1** (Appendix A). The accurate masses of these products (*m*/*z* 407.1781) correspond to a molecular weight MH^+^ equivalent to two radicals (**αS^•^** and/or **αC^•^**) (see Table 3). In comparison, eight dimeric compounds derived starting from compound **2** with MH^+^
*m*/*z* 379.1468, equivalent to two radicals (**αS^•^** and/or **αC^•^**) (Appendix A). In all cases, the fragmentation patterns are not diagnostic, all of them showing an initial fragmentation step with loss of CH_3_NH_2_. 

Further structural information may be obtained of all these dimers by taking into consideration the relative ratio of CH_3_S^•^ adducts and from the analysis of potential diasteroisomers (Table 3 and Table 4). For example, the dimer αS(2)—αS(2) is expected to be the most favorable formed since the α**S(2)^•^** radical is in a higher concentration. Figure 5 and Figure 6 show that αS(2)—αS(2) has four stereocenters, two are from the starting material fixed at the S configuration whereas two new stereocenters generated from the self-termination can be *R* or *S*. In a total of four products, two of them are identical, therefore we expect to have *SSSS*, *SRSS,* and *SRRS* diastereoisomers. The α**S(2)^•^** radical is likely to combine also with **αS(1)^•^**, **αC(1)^•^**, and **αC(2)^•^.** In analogy, the αS(2)—αC(1) is expected to be formed as four diastereoisomers (Figure 5), whereas both αS(2)—αS(1) and αS(2)—αC(2) afford only two *SSS* and *SRS* isomers (Figure 5 and Figure 6). Due to the relative high concentration of the **αS(1)^•^** radical in the experiment with compound **1**, a combination of products of **αS(1)^•^** with **αS(1)^•^**, **αC(1)^•^**, and **αC(2)^•^** are also expected (Table 3—total four diastereoisomers).

## 3. Materials and Methods 

The synthetic procedure of methionine derivative **1** and cysteine-methylated derivative **2** was described in the Supporting Information of Ref. [10].

### 3.1. Pulse Radiolysis

The pulse radiolysis experiments were performed with the LAE-10 linear accelerator at the Institute of Nuclear Chemistry and Technology in Warsaw, Poland, with a typical electron pulse length of 10 ns and 10 MeV of energy. A detailed description of the experimental setup has been given elsewhere along with basic details of the equipment and its data collection system [28,29]. The 1 kW UV-enhanced xenon arc lamp (Oriel Instruments, Stratford, CT, USA) was applied as a monitoring light source. The respective wavelengths were selected by an MSH 301 monochromator (Lot Oriel Gruppe, Darmstadt, Germany) with resolution 2.4 nm. The intensity of the analyzing light was measured by means of a PMT R955 (Hamamatsu). The signals from the detector were digitized using a Le Croy WaveSurfer 104MXs-B (1 GHz, 10 GS/s) oscilloscope (Chestnut Ridge, NY, United States) and then sent to a PC for further processing. A water filter was used to eliminate near IR wavelengths. 

Absorbed doses per pulse were on the order of 11 Gy (1 Gy = 1 J kg^−1^). Experiments were performed with a continuous flow of sample solutions using a standard quartz cell with optical length 1 cm at room temperature (~22 °C). Solutions were purged for at least 20 min per 250 mL sample with N_2_O before pulse irradiation. The *G*-values were calculated from the Schuler formula (Equation (3)) [30]: (3)$$G\left(S^{•}\right)=0.539+0.307\frac{\sqrt{19.6\left[S\right]}}{1+\sqrt{19.6\left[S\right]}}$$
where [S] is the HO^●^-scavenger concentration and with respect to the current work, the concentration of compounds **1** and **2**. This form of the Schuler formula gives *G*(S^●^) in units of μmol J^−1^, and, with respect to the current work, [S] = 0.2 mM gives *G*(S^●^) = 0.557 μmol J^−1^ where (S^●^) corresponds to all radicals formed in the system. The dosimetry was based on N_2_O-saturated solutions of 10^−2^ M KSCN, which, following radiolysis, produces (SCN)_2_^●−^ radicals that have a molar absorption coefficient of 7580 M^−1^ cm^−1^ at λ = 472 nm and are produced with a yield of *G* = 0.635 µmol J^−1^ from Equation (3) [15].

### 3.2. Spectral Resolutions of Transient Absorption Spectra

Optical spectra, monitored at various time delays following the electron pulse, were resolved into specific components (representing individual transients) by linear regression according to the following equation
(4)$$G\varepsilon \left(\lambda_{i}\right)=\underset{j}{\sum }\varepsilon_{j}\left(\lambda_{i}\right)G_{j}$$
where ε_j_ is the molar absorption coefficient of the jth species and the regression parameters, *G*_j_, are equal to the radiation-chemical yield of the jth species. The sum in Equation (4) is over all radical species present. For any particular time-delay of an experiment, the regression analysis included equations such as Equation (4) for each λ_i_ under consideration. Thus, the spectral resolutions were made using Equation (4) by fitting the reference spectra to the observed transient spectra, transformed from OD(λ) to *G*ε(λ) using the dosimetry described above. Further details of this method were described elsewhere [12,31]. 

The reference spectra of these transients were previously collected and applied in the spectral resolutions (cf. Appendix A) [12,31]. The molar absorption coefficients of the relevant transients, that will be further identified below, are provided in the following: **HOS^●^**, λ_max_ = 340 nm and ε_340_ = 3400 M^−1^ cm^−1^; **αC(1)^●^**, λ_max_ = 270 nm and ε_270_ = 6200 M^−1^ cm^−1^ and λ_max_= 370 nm and ε_370_ = 1800 M^−1^ cm^−1^; **αC(2)^●^**, λ_max_ = 340 nm and ε_340_ = 2000 M^−1^ cm^−1^**, αS^●^**, λ_max_ = 290 nm and ε_290_ = 3000 M^−1^ cm^−1^; **SS^●+^**, λ_max_ = 480 nm and ε_480_ = 6880 M^−1^ cm^−1^; **SN^●^**, λ_max_ = 390 nm and ε_390_ = 4500 M^−1^ cm^−1^, **SO^●+^**, λ_max_ = 400 nm and ε_400_ = 3000 M^−1^ cm^−1^. 

### 3.3. Steady-State γ-Radiolysis

Irradiations were performed at room temperature using a ^60^Co-Gammacell at different dose rates. The exact absorbed radiation dose was determined with the Fricke chemical dosimeter, by taking *G*(Fe^3+^) = 1.61 μmol J^−1^ [32]. 

### 3.4. LC–MS/MS Measurements

The LC–MS measurements were carried out using a liquid chromatography Thermo Scientific/Dionex Ultimate 3000 system equipped with C18 reversed-phase analytical column (2.6 μm, 2.1 mm × 100 mm, Thermo-Scientific). The LC method employed a binary gradient of acetonitrile and water with 0.1% (*v*/*v*) formic acid. Separation was achieved with a gradient of 7–60% of acetonitrile at a flow rate of 0.3 mL/min for 42 min. This UHPLC system was coupled to a hybrid QTOF mass spectrometer (Impact HD, Bruker). The ions were generated by electrospray ionization (ESI) in positive mode. MS/MS fragmentation mass spectra were produced by collisions (CID, collision-induced dissociation) with nitrogen gas in the Q2 section of the spectrometer.

## 4. Conclusions

We report experimental details on the reactions of HO^•^ with the methionine derivative **1**, as the simplest model peptide backbone, and with its congener S-methyl-cysteine derivative **2**, in order to deepen mechanistically the influence of the thioether group distance from the peptide backbone. We demonstrated that the fate of the initial one-electron oxidation at sulfur is quite different in these two cases. The pulse radiolysis experiments indicated that the one-electron oxidation at sulfur (S^•+^) occurs similarly in the two substrates, as well as the subsequent deprotonation step with the formation of **αS^•^.** These experiments were able to differentiate the formation of intramolecular sulfur–nitrogen three-electron bonded radicals (**SN(1)^•^** and/or **SN(2)^•^**), affording the C_α_-centered radicals (**αC(1)^•^** and/or **αC(2)^•^**) for methionine derivative **1** (cf. Figure 4), with the formation of intramolecular sulfur–oxygen three-electron bonded radical cations (**SO**^•**+**^), affording solely the **αC(2)^•^**) for the S-methyl-cysteine derivative **2** (Figure 5). Moreover, the product studies based on LC–MS and high-resolution MS/MS analysis from the continuous γ-radiolysis experiments allowed us to determine the deprotonation of S^•+^ intermediates giving rise to the **αS(2)^•^**/**αS(1)^•^** ratio, clearly indicating that the deprotonation of S^•+^ to form **αS(2)^•^** is more efficient for **2** than **1**. 

## Data Availability

All data are displayed in the manuscript and Appendix A.

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
