# Peer review of "Evaluation of Hydroxyl Radical Reactivity by Thioether Group Proximity in Model Peptide Backbone: Methionine versus S-Methyl-Cysteine"

_ijms, 2022, doi:10.3390/ijms23126550_

Round 1
Reviewer 1 Report
The authors presented a comparable study of the reactions between HO radical and two different peptide backbones, including methionine derivative and S-methyl-cysteine derivative. They used both radiolysis experiment, LC-MS and high resolution MS/MS to characterize and quantify the transient species in the one electron oxidation reaction. However, there are several issues in the writing and scientific soundness.
Major points to address:
1. The authors wanted to establish the impact of thioether group distance on the reaction pathways between OH radical and peptide backbone. However, they only selected two model compounds, which is not adequate. The authors may want to conduct a more systematical study on the thioether group distance effect.
2. The authors didn't explain why the pH needs to be 7, while pH constitute a significant factor governing the reaction pathways and intermediate species.
3. I was wondering if the authors could explain why they use N2O saturated solution as the reaction medium.
Overall, I recommend this manuscript to be published after major revisions.
Reviewer 2 Report
The work of K. Bobrowski and B. Marciniak et al. follows their recent publication in Int. J. Mol. Sci. 2021, 22, 4773. Similar figures and curves (figure 1 and scheme 4 vs. figure 1 and scheme 3 in Int. J. Mol. Sci. 2021) could be found with 2-acetamido-N-methyl-4-(methylsulfanyl)butanamide (CAS 24847-32-3) or (2-acetamido-N-methyl-4-(methylthio)butanamide ?) in this publication and N-acetyl methionine methyl ester (CAS 7451-74-3) in Int. J. Mol. Sci. 2021.
It would be nice if the authors gave the name and supplier of their molecules in section 3 "Materials and Methods" instead of using methionine (Met) derivative 1 and cysteine-methylated (Cys-Me) derivative 2.
The evidence of a specific absorption at 330 nm for HOS· is convincing, but, on the other hand, I am much less convinced by the specific absorption of the species C1·, C2·, SN1·, and SN2· between 300-500 nm (figure S1). Only the authors' reference (Int. J. Mol. Sci. 2021, 22, 4773) is quoted but in this ref N-acetyl methionine methyl ester was used.
Is it possible for UV to differentiate and quantify precisely these species? Furthermore, these species could not be produced in the previous publication.
For example, I was surprised to see the formation of a relatively high concentration C2· compared to C1· (figure 2 and Table 1). How could the authors could explain the higher or similar formation of C2· vs. C1· although the concentration of intermediates (SN·) is low? The higher relative stability of C1· vs. C2· should be considered. Modelisation of these intermediates could be helpful.
Is it possible for UV to differentiate and quantify precisely these species?
I agree with the authors that C1· could not be formed via SN· with compound 2. However, I was surprised that C1· was not formed by direct reaction with HO·. In line 357, the authors write that the direct formation "seems to be also unimportant". Why? If this is so, why is the direct formation of C1· mentioned for compound 1?
The work of K. Bobrowski and B. Marciniak et al. deserves publication, but a discussion on the attribution of the different absorption spectra of the short-lived radical species is needed.
Replace the comma with dots in the figures on the axes of figure 2.
Round 2
Reviewer 1 Report
None.
Reviewer 2 Report
Corrections have been made.